# Scaffolds for Cartilage Tissue Engineering from a Blend of Polyethersulfone and Polyurethane Polymers

**DOI:** 10.3390/molecules28073195

**Published:** 2023-04-03

**Authors:** Monika Wasyłeczko, Elżbieta Remiszewska, Wioleta Sikorska, Judyta Dulnik, Andrzej Chwojnowski

**Affiliations:** 1Nalecz Institute of Biocybernetics and Biomedical Engineering, Polish Academy of Sciences, Trojdena 4, 02-109 Warsaw, Polandachwojnowski@ibib.waw.pl (A.C.); 2Institute of Fundamental Technological Research Polish Academy of Sciences, Laboratory of Polymers and Biomaterials, Pawińskiego 5b, 02-106 Warsaw, Poland

**Keywords:** articular cartilage, cartilage tissue engineering, hydrolysis process, materials for scaffolds, partly degradable scaffolds, polyethersulfone–polyurethane scaffolds, polyurethane degradation, regenerative medicine, scaffold requirements, tissue engineering

## Abstract

In recent years, one of the main goals of cartilage tissue engineering has been to find appropriate scaffolds for hyaline cartilage regeneration, which could serve as a matrix for chondrocytes or stem cell cultures. The study presents three types of scaffolds obtained from a blend of polyethersulfone (PES) and polyurethane (PUR) by a combination of wet-phase inversion and salt-leaching methods. The nonwovens made of gelatin and sodium chloride (NaCl) were used as precursors of macropores. Thus, obtained membranes were characterized by a suitable structure. The top layers were perforated, with pores over 20 µm, which allows cells to enter the membrane. The use of a nonwoven made it possible to develop a three-dimensional network of interconnected macropores that is required for cell activity and mobility. Examination of wettability (contact angle, swelling ratio) showed a hydrophilic nature of scaffolds. The mechanical test showed that the scaffolds were suitable for knee joint applications (stress above 10 MPa). Next, the scaffolds underwent a degradation study in simulated body fluid (SBF). Weight loss after four weeks and changes in structure were assessed using scanning electron microscopy (SEM) and MeMoExplorer Software, a program that estimates the size of pores. The porosity measurements after degradation confirmed an increase in pore size, as expected. Hydrolysis was confirmed by Fourier-transform infrared spectroscopy (FT-IR) analysis, where the disappearance of ester bonds at about 1730 cm^−1^ wavelength is noticeable after degradation. The obtained results showed that the scaffolds meet the requirements for cartilage tissue engineering membranes and should undergo further testing on an animal model.

## 1. Introduction

Membranes are increasingly used in regenerative medicine, including tissue engineering (TE). A development of new membranes permits the replacement of transplants or provides new alternative methods allowing the treatment of damaged tissues or even diseases such as diabetes [1,2,3]. For example, flat membranes can be used for skin regeneration [4,5], the spatial form of membranes (scaffolds) are used for cell support during cultivation [6,7,8,9], and hollow fiber membranes (HFMs) are used in dialysis [10]. Membranes are also used to encapsulate active ingredients/cells in a drug delivery system (DDS) [1,11,12].

Tissue damage is a serious problem affecting many people of all ages. It can be caused by trauma, an unhealthy lifestyle, degenerative changes, or inflammatory diseases [2,13,14,15,16,17]. Some tissues, such as hyaline cartilage, due to its lack of vascularization and innervations, show limited regenerative capacity [18,19,20]. Articular cartilage is necessary to ensure proper movement [20,21]. Therefore, cartilage damage requires medical intervention, such as the microfracture (MF) method, or cell-based approaches, for example, autologous chondrocyte implantation (ACI). None of these methods produce sufficient results because they promote regeneration to fibrocartilage tissue which has inferior biomechanical properties and is susceptible to further damage [19,22,23,24]. The MF method involves stimulating the subchondral bone marrow, where a blood clot fills the defect [19,23,24,25]. The ACI method enables the implantation of the patient’s articular chondrocytes (ACs) in place of the cartilage lesions. Cells are either delivered directly to the damaged site or are delivered in a scaffold that acts as a temporary matrix [19,23,26,27]. Studies show better results for ACI compared to MF [23,28]. Doctors and scientists are still looking for an effective method to regenerate cartilage. A promising solution for restoring hyaline cartilage is to combine well-known methods such as the MF technique with scaffolds, which will provide the right conditions for mesenchymal stem cells [8,21,27,29]. Therefore, scaffolds tailored for cartilage needs deserve special attention.

The architecture of scaffolds suitable for this application should be characterized by a strongly developed three-dimensional spatial structure with a network of interconnected pores of the right size. It is important since such construction mimics the environment of a living organism (native tissue) much better, compared to two-dimensional membranes [30,31,32,33]. It affects the settlement and migration of cells, as well as their products, which consequently leads to cartilage regeneration. It requires the presence of micropores, which will be responsible for both supplying nutrients and oxygen to cells, and also for removing metabolism products outside the membrane [8,9,34,35]. Moreover, scaffolds should be characterized by biocompatibility and degradability. This can be achieved by selecting, carefully, the method of preparation, materials, and additives, such as pore precursors [9,35,36]. Depending on the application (cell selection), a scaffold with the appropriate pore size should be selected. Namely, stem cells are required correspondingly to their size, with larger pores inside the membrane [8,9,37,38]. What is more, the scaffold’s parameters such as mechanical strength, stiffness, and flexibility are essential during cell cultivation and after implantation into the body [30,31,32,35,39]. A desirable feature of scaffolds is controlled degradability in conditions that are present in the human body. The degradation time should not be too long or short and be proportional for tissue regeneration [35,40,41].

By properly selecting the scaffold material, both the degradation time, as well as its mechanical properties, can be easily controlled [6,9,27,34,35,41,42,43,44]. Natural polymers such as sugar and protein compounds such as collagen (COL) [45,46], hyaluronic acid (HA) [47], chondroitin sulfate (CS) [48,49], and fibrin [50] could be used to develop materials for cartilage regeneration. Natural materials are characterized by high bioactivity and biocompatibility, and they have properties similar to those of native tissues. Although they have many advantages, they are not the easiest materials for scaffold production. They are usually sensitive to elevated temperatures, pressure, and varying pH. Membranes made solely of these polymers have poor mechanical strength, and their rapid hydrolysis (sensitivity to an aquatic environment) causes a loss of the proper structure of the scaffold [27,30,32,35,44,45]. Synthetic polymers, compared to natural polymers, have better mechanical resistance and durability under various conditions. The most commonly used synthetic polymers in scaffold development are polylactic acid (PLA) [51,52], polycaprolactone (PCL) [53,54], polyurethane (PUR) [55,56], and polyethersulfone (PES) [7,57,58,59]. Due to their good mechanical, physical, and chemical properties, they can be used to produce various scaffold shapes using different techniques. Those which are biodegradable break down into components that are non-toxic for the host and metabolized in the body. In addition, their mechanical properties and degradation time can be controlled by combining two or more polymers (as copolymers or blends) [29,32,44,60,61,62,63,64,65,66]. It is worth noting that some synthetic materials do not have adequate biological properties, and sometimes degradation products, such as acids, can cause side effects on the host organism, for example, induce an inflammatory response [32,67,68]. Considering the advantages of synthetic and natural polymers, hybrid membranes with good biological and mechanical properties can be obtained, which is the current direction of scientists [23,27,30,41,67,69,70]. Moreover, noteworthy are membranes made of a blend of polymers, one of which is biodegradable. The disappearance of one of them that happens over time improves fluid transport and separation properties. For instance, increasing porosity/pore size will affect the release of space for regeneration, while the more stable polymer will preserve the scaffold skeleton [42,43,60,62,64,70].

Many scaffolds are undergoing preclinical and clinical tests. Commercial scaffolds used in cartilage regeneration are generally made of natural polymers (mainly collagen). Due to the mentioned disadvantages of natural materials, these scaffolds do not meet the necessary requirements (low mechanical stability and rapid hydrolysis). They quickly lose their structure because they turn into a gel-like form. This leads to the regenerated non-valuable fibrocartilage, which is susceptible to further damage [23,30,35,41,44,71].

This study aimed to obtain partially degradable scaffolds for cartilage tissue engineering. Three membranes were obtained from a blend of PES and synthesized PUR polymers, in different weight ratios. The PES–PUR blend was chosen because of the biocompatibility of PES and PUR (important for biomedical applications), the good solubility of both polymers in the same solvents (important for manufacturing), and the presence of ester bonds in PUR (hydrolysis in physiological fluid). Water-soluble substitutes were used to obtain micro- and macropores in the scaffold (adequate structure of membranes). The scaffolds were prepared using a combination of wet phase separation and salt leaching techniques. In this work, the possibility of partial degradation using simulated body fluid (SBF) at a temperature of 36 ± 2 °C (in vitro degradation) was evaluated. The mechanical, morphological, and physical properties of the scaffolds were determined, among others, by scanning electron microscope (SEM), Fourier-transform infrared spectroscopy (FT-IR), and contact angle examination.

## 2. Results and Discussion

### 2.1. Morphology Characterization of the Membranes before Degradation

Scaffolds as a temporary cellular environment allow cells to function as in a native tissue. However, they should meet a certain set of requirements, including the suitable architecture, or physicochemical and biological properties. This is essential for proper cell growth, which involves adequate tissue regeneration [30,31,32,35,39,43]. The morphology of the membranes was characterized by SEM, which made it possible to obtain information regarding both the surface and cross-section of each membrane.

The SEM images of M1-M5 membranes are shown in Figure 1 and Figure 2. The reference membranes M1–M2 were made by the phase inversion method, without the addition of NaCl or nonwoven gelatin. Their structure was compact, and this was particularly noticeable for M2. They were made to observe pure PES and PUR polymers without adding macropores generators. In contrast, the M3–M5 (Figure 2) scaffolds adequately met the conditions necessary for cell culture. They were made using a macropore generator—NaCl and gelatin nonwoven. The top layer was perforated, allowing cells to enter the membrane. Their cross-section was characterized by a unique spatial structure formed by an interconnected network of pores [7,36,62]. Their main task, in addition to keeping cells in space, is to provide an environment as similar as possible to that naturally occurring in the human body. It is expected that using a scaffold of proper architecture will ensure that the cells cultured within the scaffold will have the right shape, and their metabolism and vital functions will be normal. The bottom layer, on the other hand, was compact, which should prevent cells from escaping. In addition, micropores were visible in the cross-section of a membrane formed by an addition of PVP 10 kDa and Pluronic F127 precursors of pores (Figure 2: M3CSA, M4CSA). They affect oxygen and nutrient permeation and metabolite removal [8,9,34,35].

### 2.2. Degradation Studies

#### 2.2.1. SEM Imaging and pH Measurement during and after Degradation

In vivo experiments using an animal model are not always available to elucidate the mechanism of membrane degradation. Instead, various in vitro experiments have been designed to simulate membrane’s degradation in vivo. In this work, hydrolytic degradation was carried out using three fluids. The study investigated how the degradation of the obtained membranes proceeds. Whether it is possible and how it proceeds over four weeks in different liquids.

Degradation was performed in 1 M NaOH, Hank’s balanced salt solution (HBSS), and a simulated body fluid (SBF). It was carried out in an incubator, at 36 ± 1 °C. Membrane weight tests were performed weekly (Figure 3, Figure 4 and Figure 5, while FT-IR (Section 2.7), pH (Table 1), and SEM observations were performed after 2 and 4 weeks. The SEM photomicrographs of the membranes after 2 and 4 weeks of degradation are available in the Appendix A. For the facilitation of discussion, the abbreviations of MxYy were used, where x denoted the membrane number, Y denoted the degradation fluid (where Y for SBF was B; for HBSS, H; and for NaOH, N), and y denoted the number of the week. For example, M1B2 indicates the degradation of the membrane M1 in SBF after two weeks.

After 2 weeks, cracks were noticeable on the membranes, with the exception for the M5H2 and M5B2 scaffolds, where no changes were found. After 4 weeks, changes were visible in all cases, especially for M2N4, M3N4, and M5N4, where destruction had occurred almost completely. Small fragments and crumbs remained of these scaffolds. Even the membranes made of PES (M1) and the M4 scaffold, with twice the advantage of PES over PUR, were very brittle after 4 weeks, and they were prone to fracture.

As can be seen in Table 1, the pH changed during and after degradation. No high decreases or increases in pH were noted, which could affect the organism [72,73,74].

During SEM observations, attention was focused on the presence of spheroids on the membranes after 4 weeks of degradation in SBF and HBSS, where by far the highest amount was observed for the M4 scaffold (Figure 6). From the literature review, it was noted that these spheroids could be a result of apatite formation on the membranes [75,76,77,78].

#### 2.2.2. Weight Loss during Degradation

Scaffold weights were measured each week. After removal from the medium, the samples were properly rinsed in redistilled water and dried at 30 °C. The graphs (Figure 4, Figure 5 and Figure 6) show the percentage of mass loss of the membranes during four weeks of degradation in each media.

For each of the membranes, a weight loss during degradation was noted. The results for M2 membrane, made of PES, were far from expected, as even a mass loss of up to 12.7% ± 0.0002 in SBF was noted for it, while for HBSS it was 4.9% ± 0. Moreover, the M4 scaffold, in which the weight ratio of PES was twice that of PUR, showed significant weight loss: 29.1% ± 0.001 in SBF and 13.7% ± 0.001 in HBSS. Among the M3–M5 scaffolds, the lowest percentage of mass loss was for M3—12% ± 0.001 in SBF and 11.5% ± 0.0001 in HBSS.

The highest percentage of mass lost during degradation was recorded for scaffold M5, in which there was a two-times weight ratio advantage of PUR over PES. In the SBF medium, it was 32.7% ± 0.0007, while in HBSS, it was 16.1% ± 0.0004. In a previous study in a Ph.D. thesis by Tomasz Jakutowicz, the absence of a PES scaffold was observed during animal model testing after 6 months of implantation into the joint [79]. The study observed PES degradation in SBF and HBSS, where there is no information on that topic in the literature. The degradation time in SBF, which is most closely related in composition to blood plasma, showed different degradation rates for scaffolds of different material composition. This showed that with the proper selection of the weight ratio of polymers, the degradation time can be controlled [42,43].

The situation was different for the hydrolysis experiment in 1 M NaOH. The highest percentage of sample mass loss caused by hydrolysis after 4 weeks was observed for the M5 scaffold, and it was 52.3% ± 0.0014, whereas for M1, no change in mass was observed. For M2, M3, and M4 membranes, the mass loss was 44.6% ± 0.0034, 45.8 ± 0.0029, and 44.7 ± 0.0172, respectively. M3 and M5 scaffolds were completely deconstructed after 4 weeks. Small pieces and crumbs remained of the membranes. Changes in the membranes were noticeable in the case of each material. After 4 weeks, membranes were brittle and friable, even in the case of M1.

Hydrolysis and biodegradation of capillary membranes made of a mixture of PUR and polysulfone (PSf) have been performed in previous works using 1 M NaOH [60,63] and *E. coli* [64]. The (bio)degradability of synthesized PUR was demonstrated. In this work, in vitro biodegradation was presented using simulated physiological fluids, which have not been used before relative to PES, and this kind of PUR materials, which shows the novelty of this research. Further research should be conducted in this direction.

### 2.3. Estimate of Pores in Scaffolds by Computer Analysis of SEM Images

#### 2.3.1. Pore Distributions and the Total Aera of Pores for Scaffolds before Degradation

It is impossible to determine exactly what numerical changes occur in pore size by evaluating SEM images analyzed by comparing two pictures with the human eye. It is not a precise method. Therefore, to more accurately assess changes in pore size, SEM micrographs were analyzed using MeMoExplorer ™ Software [80].

Figure 7 presents the area of pores in eight size classes for the cross-section (CS) and the top layer (TL) of scaffolds. The results show the average percentage of appropriate pore size to the whole SEM photomicrographs size.

In each scaffold, the largest number of pores were of those with an area over 300 µm^2^. This was noticeable in the case of cross-section (CS) and the top layer (TP). The highest percentage of these pores was recorded for the top layer of M3, 21.54% ± 10.89, and for the cross-section of M4, 20.66% ± 11.22, while the top layer for M4 was 20.73% ± 9.49. In contrast, the result for cross-section for M3 was lower than M4 at 17.08% ± 5.61. The lowest percentage of pores above 300 µm^2^ was recorded for M5, 16.51% ± 7.72 for cross-section, and 8.58% ± 5.46 for the top layer. The 150–300 µm^2^ range of pores were relatively frequent at 2.48–4.37%, and the 20–80 µm^2^ pores were in the range of 4.1–2.7%. The pores with an area of 0–3 µm^2^ turned out to be the least common, ranging from 0.09–0.51%.

#### 2.3.2. Total Area of Pores after Degradation

Next, the data on the total area of pores for scaffolds before and after degradation (Figure 8, Figure 9 and Figure 10) were calculated. The highest percentage of the total area of pores before degradation was obtained for the M4 scaffold cross-section (32.33% ± 2.56), while for the surface, it was recorded for M3 (30.88% ± 9.04). The total area of pores for M4 on the top layer was slightly lower than M3 (29.18% ± 5.94), while the percentage for the M3 cross-section was 30.38% ± 3.76. The lowest percentage of the total area of pores was for M5: for the surface, it was 21.20% ± 4.69, and for the cross-section, it was 29.14% ± 5.42.

The largest change in the area of pores was recorded during hydrolysis in 1 M NaOH. For the cross-section of the M4 scaffold, 55.79% ± 17.40 (23.46% increase) was obtained, and for the top layer, 47.96% ± 12.47 (18.78% increase). Unfortunately, a comparison of the M4 scaffold against the M3 and M5 membranes was not obtained, because of their destruction. The difference in the change in the number of pores in the surface and cross-section is shown in Table 2.

The highest difference in the total area of pores was noticed for scaffold M4 in HBSS fluid for the cross-section and the top layer. The situation was different for the SBF medium, where it was observed for scaffold M5 (Table 2). The lowest difference was for the cross-section of the M4 scaffold in SBF (5.80) and the top layer of the M5 scaffold in HBSS (0.89).

A comparison of the area of pores > 300 um in the cross-section was made before and after degradation in physiological fluids (Table 3). The difference was noticeable mainly in the M5 scaffold in SBF. The increase in pores area was twofold. This information suggests that space was freed up in the scaffolds as a result of increased surface area in the largest pores. The difference was mainly noticeable for the M5 scaffold in SBF. The increase in pore area was twofold. In contrast, it was the smallest for the M4 scaffold in SBF [43].

### 2.4. Porosity of Membranes

The study also determined the porosity of the membranes before and after degradation. The diagram (Figure 11) shows the porosity of the membranes before degradation. The calculations were made using Equation (2). The highest porosity was recorded for scaffold M4 (87.37% ± 2.74), while the lowest porosity was obtained for membrane M2, which was only 6.74 ± 13.54.

Figure 12 shows the porosity of the membranes after degradation and the difference (Diff) that was measured before and after degradation. The highest porosity was recorded for the M4 scaffold in NaOH (94.99% ± 0.88), where the difference was 7.62%. The largest difference in porosity was calculated for the M2 membrane in NaOH, where the porosity was 80.73% ± 4.45. The increase in porosity was 53.99%. There was a slight increase in porosity for M3H4 and M4B4, by 0.39% and 0.75%, respectively. Changes in pore size after degradation were shown in porosity and SEM image analysis. This is very important information, as it indicates that during degradation, space is released for newly forming tissue [60,62,64,70].

### 2.5. Wettability of Membranes

An essential feature of cellular scaffolds is their hydrophilicity. This parameter determines the possibility of migration of body fluids, water, and nutrients into the interior of the scaffolds, and it thus defines the vital functions of the cells developing in them [81,82,83]. PUR and PES polymers, due to their chemical structure, are hydrophobic. Because of a concern that the scaffolds would also be hydrophobic, it was decided to test their swelling ratio using PBS and the contact angle using deionized water. The M1 and M2 reference membranes and M3, M4 and M5 scaffolds were investigated, which from the point of view of cell culture would meet the requirements for cell culture.

#### 2.5.1. Swelling Ratio

Scaffolds M3–M5 showed very good saturation with hydrophilic substance PBS in contrast to reference membranes M1 and M2 (Figure 13).

This shows that the additives used during the preparation of the scaffolds (pore precursors) not only increased the size and number of pores and porosity, but also the property of the scaffolds. The number of pores and their size increases the absorbency of the samples, which has an impact on degradation. The highest swelling ratio was observed in scaffold M4: 506.02% ± 60.31 after 3 h and 550.70% ± 57.03 after 24 h. Scaffold M3 (494% ± 77.83 after 3 h and 502.38% ± 62.70 after 24 h) showed similar absorbability. The M5 scaffold showed a swelling ratio of 263.81% ± 36.46 after 3 h and 293.43% ± 28.08 after 24 h. The lowest saturation value came out for reference membranes M1 and M2 (57.11% ± 10.35 and 82.44% ± 11.85, respectively, after 3 h), which were obtained without the use of non-woven gelatin and NaCl. In these tests, the results after 3 h and 24 h were similar. These studies confirm the correlation between scaffold morphology and saturation, which also has an impact on scaffold degradation. Namely, the porosity and swelling ratio were the highest for M4, which also resulted in its increased porosity and size of pores after degradation (Figure 11, Figure 12 and Figure 13) [81,82,83].

#### 2.5.2. Contact Angle

The nature of the membrane surface was studied by measuring the contact angle. Figure 14 presents a diagram showing the average wetting angle (Ave) for the membranes along with the standard deviation (SD). In addition, photos of the membrane surface with a water droplet are presented above the diagram. The contact angles of all the membranes were below 62°, which means that each membrane had a hydrophilic surface.

For M1 and M2 membranes, the wetting angle was 54.45° ± 1.40 and 55.67 ± 1.12, respectively. According to the literature, the value of the pristine PES membrane is 60–65 [84,85]. In the study, the M1 membrane had a smaller contact angle because using the pore precursors (poly(vinylpyrrolidone) and Pluronic) improves hydrophilicity. This is often used to refine the hydrophilic properties of membranes [8,85]. The higher contact angle was obtained for M5 (61.84 ± 0.81). The most hydrophilic scaffold proved to be M4, 51.26° ± 1.06, and M3, which was 30.02 ± 1.10. These results were similar to the swelling ratio (Section 2.5.1), where the lowest outcome was for M5.

Measurements were also made for the membranes after degradation. In each case, the water droplet was soaked in very quickly (Figure 15). Adsorption of water by the scaffold determines the permeation of proteins, the improved release of nutrient ions (cell nutrition), or cell-surface adhesion. This is essential for regulating cell metabolic functions. In addition, polymer degradation is dependent on the surface properties of the polymer. Hydrolytic degradation does not occur until water enters the scaffold [86,87,88].

### 2.6. Mechanical Properties

It is important to consider the mechanical properties of scaffolds for cartilage regeneration. This is an essential parameter for the proper functioning of the membrane in the joint and should be close to the strength of the natural tissue. Young’s modulus value of human articular cartilage in tension is approximately 10 MPa [89,90]. The mechanical tests were performed for each membrane (Figure 16). All tested scaffolds showed good mechanical properties (E > 10 MPa). The reference membranes showed a very high Young’s modulus, where the porous scaffolds M3-M4 have a score 3.5 times lower, but they do not fall below 10 MPa. It means that they would certainly withstand the conditions experienced in the knee joint. The PES material showed greater strength, as proven by measurements for M1 and M4 membranes. By changing the weight ratio of PUR/PES (Table 4), the value of Young’s Modulus of membranes can be controlled. It could be seen that by increasing the PES polymer, the mechanical strength increased as well. Additionally, it was observed that the less porous membranes had better properties.

### 2.7. FT-IR Analysis

The membranes were characterized by FT-IR to study changes after 2 and 4 weeks of degradation. Figure 17 shows the spectra of the membranes before degradation. It can be seen that there were differences in the intensity of peaks corresponding to wavelengths in the 1800–1500 cm^−1^ range (Figure 17, blue square).

The difference in absorbance was observable for the peak at 1730–1740 cm^−1^, assigned to the C=O functional group derived from ester bonds. The peak had the highest absorbance for membrane M2, while it was not noticeable for M1. The intensity of the peaks decreased with PUR weight ratio (Table 4). The peaks at 1700–1630 cm^−1^ can be attributed to carbamate groups. The peaks in the range of 1600–1570 cm^−1^ represent the aromatic bands, the C-S group characteristic of the PES polymer, and the amine groups, which is occurring in PUR [85,91,92].

Figure 18 shows the spectra of the M1 membranes before and after degradation. There were no visible changes in any of the fluids after 4 weeks of degradation. It indicates that the membrane did not change chemically, only physically—the structure was more brittle, especially after degradation in NaOH.

Changes in the membranes M2–M5 after degradation were noticeable. Both disappearance and increase in the absorbance of peaks at the 1800–1600 cm^−1^ range were observed. The spectra of M2–M5 membranes before and after degradation are presented in the Appendix A. Changes were noted for the peaks at about 1730 cm^−1^ (ester bonds) and in the 1670–1620 cm^−1^ range (carbamate group).

The disappearance of the ester group was visible after two weeks for membranes M2, M4, and M5 in liquid NaOH. After four weeks, it is also noticeable for M3 in NaOH and M5 in HBSS liquid. Reductions in absorbance for the carbonyl group were observed for the M2–M5 membranes [74,76,93].

## 3. Materials and Methods

### 3.1. Materials

Chemicals used in our experiments, namely, Polyvinylpyrrolidone (PVP, M_n_ = 10 kDa), 1–Methyl-2-pyrrolidone (NMP), phosphate-buffered solution (PBS), Pluronic^®^ F127, hydrochloric acid (HCl 70%), sodium bicarbonate (NaHCO_3_), sodium azide (NaN_3_), Tris ((CH_2_OH)_3_CNH_3_), phenol red, monopotassium phosphate (KH_2_PO_4_), D-Glucose, disodium phosphate (Na₂HPO₄), and magnesium sulfate (MgSO₄), were acquired from Sigma-Aldrich (Steinheim in Germany). Sodium hydroxide (NaOH), dimethyl sulfoxide (DMSO), N, N-dimethylformamide (DMF), hexane, sodium chloride (NaCl), potassium chloride (KCl), magnesium chloride hexahydrate (MgCl_2_·6H_2_O), calcium chloride (CaCl_2_), potassium hydrogen phosphate trihydrate (K_2_HPO_4_·3H_2_O), and Sodium sulfate (Na_2_SO_4_) were purchased from POCh SA (Gliwice in Poland). Ethanol (EtOH 96%) was provided by Linegal Chemicals (Warsaw in Poland). Polyethersulfone (PES) Ultrason E2020P (Figure 19) was acquired from BASF (Ludwigshafen am Rhein in Germany).

Polyurethane (PUR) (Figure 20), with ≈90% molar content of ester bonds in the structure, was synthesized using methods from articles [60,63,64].

The gelatin (from bovine skin, Type B, ~300 g Bloom, Sigma-Aldrich) nonwovens (Figure 21 were obtained by an electrospinning method in the Institute of Fundamental Technological Research PAS, according to the literature [94,95].

### 3.2. Preparation of the Reference Membranes M1 and M2

Preparation of solutions:M1: The PES polymer was dissolved in DMF and DMSO (mixed in a ratio of 1:1) to obtain 8.6 wt.% concentration. Then, the 2.5 wt.% PVP 10 kDa and 2.5 wt.% Pluronic F127 were added with constant stirring at room temperature until a solution was achieved.M2: The PUR polymer was dissolved in DMF and DMSO (mixed in a ratio of 1:1) to obtain 8.6 wt.% concentration. Then, the 2.5 wt.% PVP 10 kDa and 2.5 wt.% Pluronic F127 were added with constant stirring at room temperature until a solution was achieved.

The M1 and M2 membranes were obtained by the phase method according to the scheme (Figure 22). An appropriately prepared solution was poured onto a glass base. Both membranes were immersed in the bath with deionized water for up to 24 h. The deionized water (18.2 MΏcm conductivity) was obtained using the arium^®^ comfort apparatus (Sartorius).

### 3.3. Preparation of the PUR-PES Scaffolds M3–M5 Using Gelatin Nonwoven

Scaffolds were obtained by combining wet phase inversion and salt leaching methods using different polymers (PES, PUR) weight ratios (Table 4). Solutions were prepared by dissolving the polymer PES, 2.5 wt.% PVP 10 kDa and 2.5 wt.% Pluronic F127 in DMF and DMSO at a ratio of 1:1. Then, PUR was then added to obtain 13.6 wt.% concentration. Three scaffolds (M3–M5) were obtained from the solutions using the appropriate weight ratios of PES and PUR, gelatin nonwovens, and NaCl salt (Figure 23). To obtain NaCl salt with suitable crystals, laboratory grinder FW177 (hemLand) and laboratory sieve with 200 µm mesh were used.

The membranes were obtained as follows: the NaCl salt was spread evenly on the cooled glass base (about 4 °C), then the gelatin nonwoven was laid down, and the appropriate polymer mixture was poured onto it. Next, 2 layers of nonwovens were arranged until they were saturated, and a second layer of membrane-forming solution was added. All layers were pressed with a Teflon roller to remove air bubbles. The received scaffolds were gelled in the bath with deionized water and ice (the temperature of the bath was about 2 °C). Gelation occurred with simultaneous salt removal. After 12 h, the gelatin was removed by immersing the membrane in a warm deionized water bath (50 °C).

### 3.4. SEM Observation

The morphology of the top and bottom layers, and cross-section of the membranes before and after degradation were characterized by scanning electron microscope (Hitachi TM–1000, Tokyo, Japan) with an accelerator voltage of 15 kV. The membranes were cut in liquid nitrogen, dried, and then coated with a 7 nm gold layer, using a sputter coater (EMITECH K550X, Warsaw, Poland).

### 3.5. Estimate of Pores in Scaffolds by MeMoExplorer™ Software

To evaluate changes in the morphology of the cross-section and top layer in the scaffolds (in pore size), selected samples were analyzed using MeMoExplorer™ software (Warsaw, Poland). This program evaluates images obtained from SEM analysis [60,62,80]. The SEM images were taken with microscope magnification of ×300 or ×500. Then, they were analyzed by software which involved contouring of pores surfaces and their evaluation with partitioning them into 8 size classes (Table 5) and calculation of total areas (porosity coefficients).

An average of 30 SEM images were taken for each sample. The received data can be processed statistically to obtain parameters such as average (Ave) and standard deviation (SD). That can be performed by using suitable software such as Origin or Microsoft Excel.

### 3.6. Degradation of Scaffolds

Scaffolds M1-M5 degradation was performed in three different fluids: in 1 M NaOH (pH = 13.50), Hank’s balanced salt solution (HBSS) (pH = 7.54), and in a simulated body fluid (SBF) (pH = 7.31). The SBF and HBSS were chosen for evaluating in vitro biocompatibility as they simulate physiological fluids. Furthermore, an SBF with ion concentrations nearly equal to those of human blood plasma [75,96] (table of ion concentrations is available in Appendix A).

The liquids were prepared in the laboratory according to the method given in the literature [75,96]. The scaffolds were cut into rectangles, which were measured (length, width, and thickness) using a caliper tool and weighed by analytical balance (MATTLER TOLEDO KA-52c, Warsaw, Poland). The shape of the membranes varied due to their different morphology. It was not possible to cut them into samples of similar size. The samples of scaffolds (*n* = 6) were immersed into plastic bins filled with 40 mL of each liquid for 4 weeks at 36 ± 2 °C in a small multi-purpose incubator CULTURA M, 70700R (Almedica, Krakow, Poland). Every week, samples were washed in deionized water, dried, and weighed, and the pH values of liquids were monitored using an electrolyte-type pH meter (METTLER TOLEDO MP225, Warsaw, Poland). The mass loss was calculated from the following equations (Equation (1)) [97]:(1)$$\mathrm{Weight}\;\mathrm{loss}=\frac{\left(M_{0}-M_{t}\right)}{M_{0}}\times 100\;\%$$
where M_0_ and M_t_ with subscript 0 and t are mass at the immersion time of 0 and t, respectively. All the values presented were the average of six samples.

### 3.7. Porosity of Membranes

The porosity of the scaffolds was determined by measuring the mass and dimensions of the scaffolds before and after hydrolysis, as described by Ho et al. [97]. It was calculated with the following formula (Equation (2)):(2)$$\mathrm{Porosity}=\frac{D_{p}-D_{\mathrm{ap}}}{D_{p}}\times 100\%$$
where D_p_ is the density of membranes, which were, respectively, PUR (1.37 g/cm^3^), PES (1.25 g/cm^3^), M3 (1.31 g/cm^3^), M4 (1.33 g/cm^3^), and M5 (1.29 g/cm^3^); D_ap_ is the apparent density (scaffold mass/apparent scaffold cube volume).

The calculations were carried out in 10 repetitions for both scaffolds before hydrolysis and 7 repetitions after hydrolysis. All data were expressed as average (Ave) ± standard deviation (SD).

### 3.8. Swelling Ratio

To calculate the saturation of the scaffolds, squares were cut from each scaffold, respectively. The samples were then weighed (W_0_) on an analytical balance and immersed in 50 mL bins filled with PBS solution (pH 7.2–7.4). The PBS was prepared by dissolving the tablets in deionized water. The beaker with the scaffolds was set aside for 3 and 24 h, respectively. After this time, the swollen samples were weighed (W_t_). The equilibrium swelling ratio (ESR) was then calculated by using the following equation (Equation (3)) [98]:(3)$$\mathrm{ESR}=\frac{W_{t}-W_{0}}{W_{0}}$$

The 7 repetitions were performed for each membrane.

### 3.9. Contact Angle Measurement

The static wetting angle of membrane surfaces was analyzed using a DSA25 goniometer, (KRŰSS GmbH). Before the measurements, the membranes were stored in a desiccator. The measurements were carried out at room temperature at 24 ± 2 °C. A 5 μL drop of deionized water was deposited on the test surface. The results were presented as the average of 10 measurements with the standard deviation (Ave ± SD).

### 3.10. Mechanical Property

The mechanical properties were tested using a materials testing machine (Tiratest 2160, VEB Thuringer Industriewerk Rauenstein, Warsaw, Poland) at 23 ± 2 °C. The sample dimensions were 20 mm × 15 mm × 1–2 mm. The 5 repetitions of dry samples were performed for each membrane.

### 3.11. FT-IR

The scaffolds were characterized by using the Fourier-transform infrared spectroscopy ALPHA II machine (Platinum- ATR, BRUKER, Warsaw, Poland) with 4 cm^−1^ resolution, averaging for 32 measurements, wavelengths from 4000 to 400 cm^−1^. The measurements were made using the OPUS 8 software. The graphs were prepared using the OriginPro 8 program.

### 3.12. Statistical Analysis

All the quantitative data were obtained from at least five samples for analysis. Results were expressed as the average ± standard deviation (Ave ± SD).

## 4. Conclusions

Currently, the scaffolds are being sought for hyaline cartilage regeneration, which should meet the relevant requirements. In this study, the three scaffolds were prepared from a mixture of biodegradable polyurethane (PUR) and polyethersulfone (PES). These scaffolds have met the appropriate requirements for cartilage regeneration. The observation with an SEM microscope proved that the internal structure of the membranes forms a network of interconnected pores, while the surface layer is perforated, which creates the possibility for cells to enter the interior. In addition, the analysis of SEM photomicrographs via MeMoExplorer ™ Software showed that the area of pores of >300 um^2^ dominated. The scaffolds showed degradation in simulated body fluid. The wettability results showed a hydrophilic nature of scaffolds, especially after degradation. Moreover, an increase in both porosity and pore surface area has been shown for scaffolds after degradation. It is necessary to create space for cellular products such as hyaline cartilage-forming protein. Adequate mechanical properties (>10 MPa) that were shown prove that these membranes will be able to withstand conditions in the knee.

Obtained membranes have shown suitable properties as scaffolds for cartilage engineering. Such membranes can be colonized by autologous/allogeneic chondrocytes (ACI method) or mesenchymal stem cells (MF technique).

## Figures and Tables

**Figure 1 molecules-28-03195-f001:**
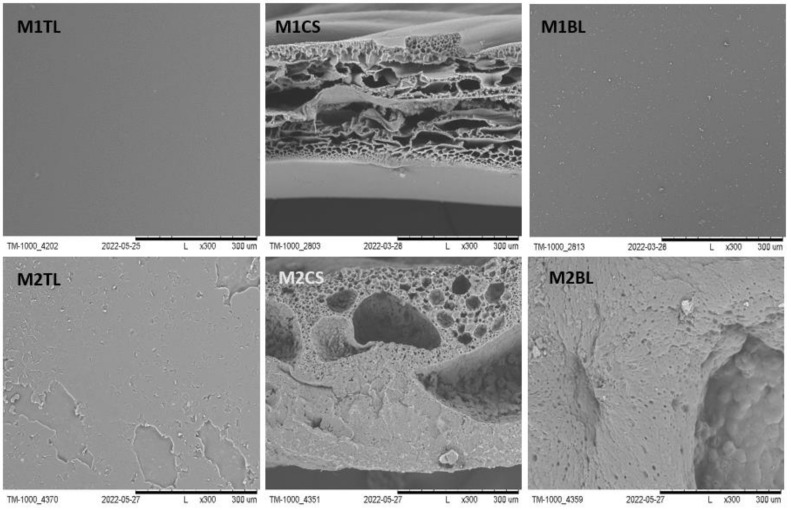
The SEM photomicrographs of the M1 and M2 membranes. TL—Top layer; CS—Cross section; BL—Bottom layer. Scale bars: 300 µm.

**Figure 2 molecules-28-03195-f002:**
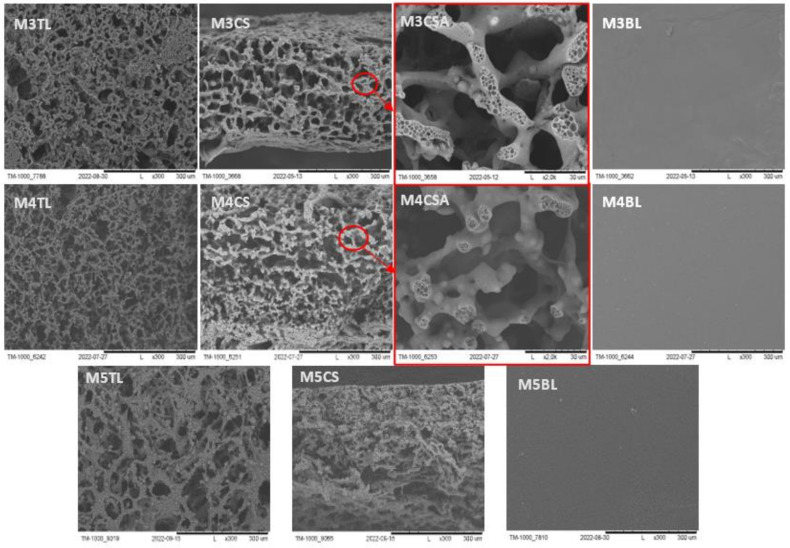
The SEM photomicrographs of the M3 and M5 scaffolds. The imagines of M3CSA and M4CSA show a magnification of the micropores that occur in the walls in cross-section (red circles). TL—Top layer; CS—Cross section; BL—Bottom layer. Scale bar: 2000 µm—M3CSA, M4CSA; 300 µm—others.

**Figure 3 molecules-28-03195-f003:**
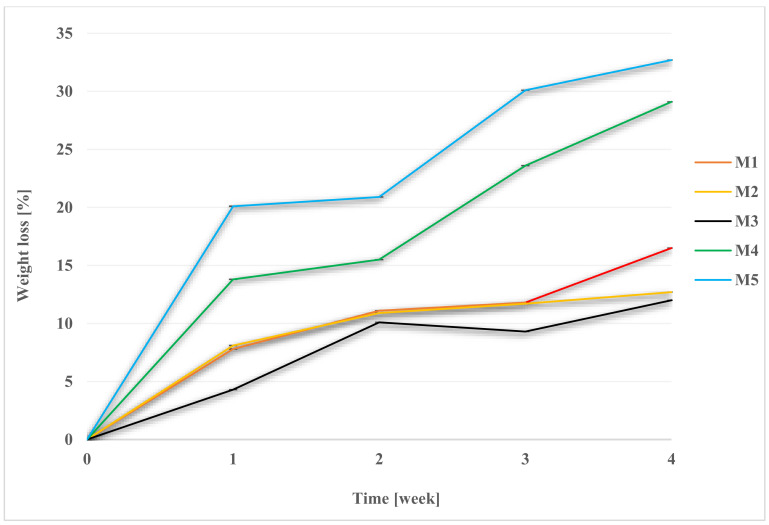
Weight loss of membranes during degradation in SBF.

**Figure 4 molecules-28-03195-f004:**
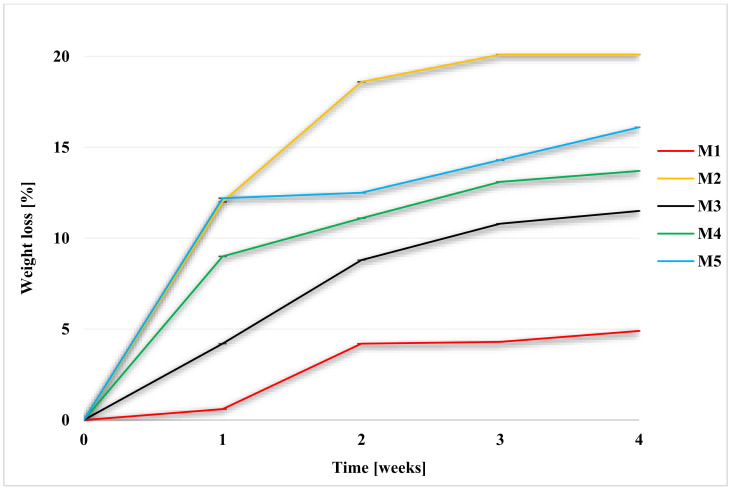
Weight loss of membranes during degradation in HBSS.

**Figure 5 molecules-28-03195-f005:**
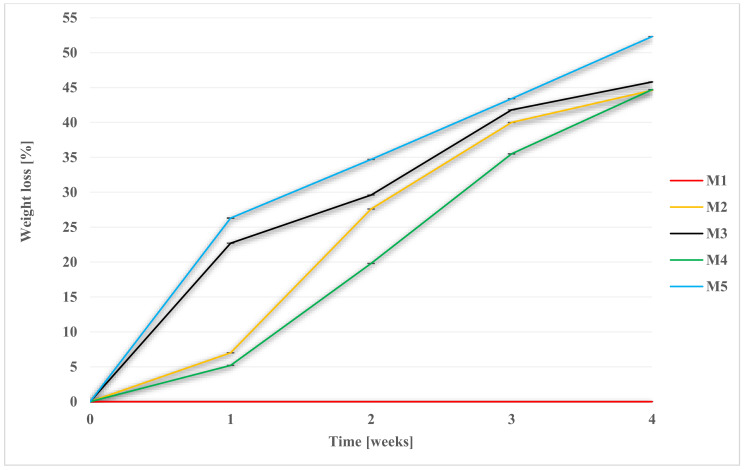
Weight loss of membranes during degradation in NaOH.

**Figure 6 molecules-28-03195-f006:**
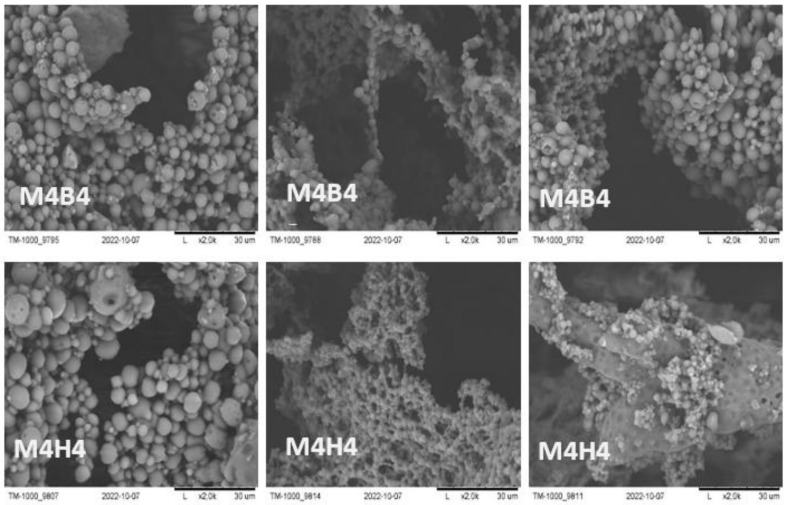
SEM micrographs of spheroids on scaffold M4 after 4 weeks degradation in SBF and HBSS.

**Figure 7 molecules-28-03195-f007:**
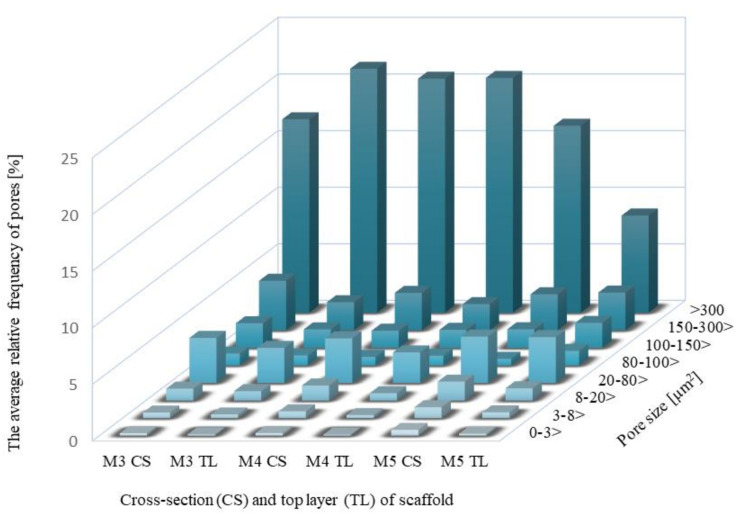
The average relative frequency of pores in eight size classes for cross-section (CS) and the top layer (TL) of scaffolds.

**Figure 8 molecules-28-03195-f008:**
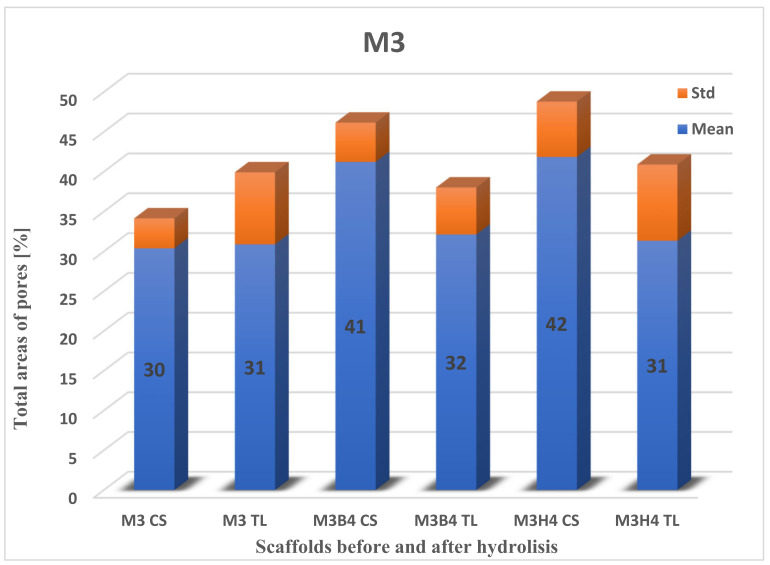
The total area of pores to the whole SEM image size for cross-section (CS) and the top layer (TL) of the M3 scaffold. B—degradation in SBF; H—degradation in HBSS.

**Figure 9 molecules-28-03195-f009:**
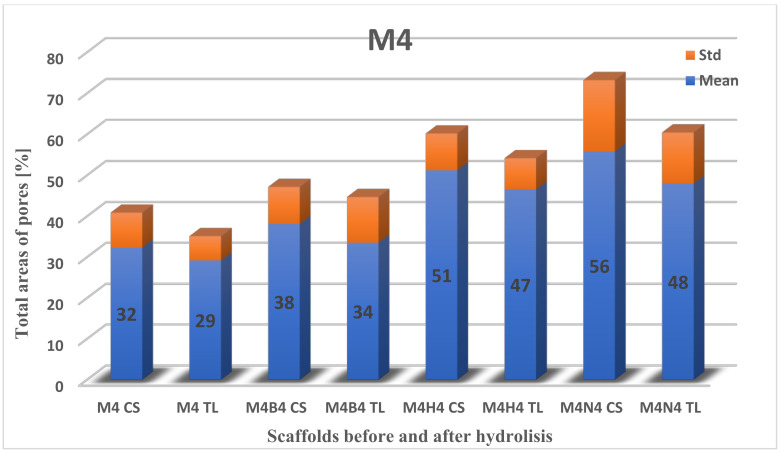
The total area of pores to the whole SEM image size for cross-section (CS) and the top layer (TL) of the M4 scaffold. B—degradation in SBF; H—degradation in HBSS; N—degradation in NaOH.

**Figure 10 molecules-28-03195-f010:**
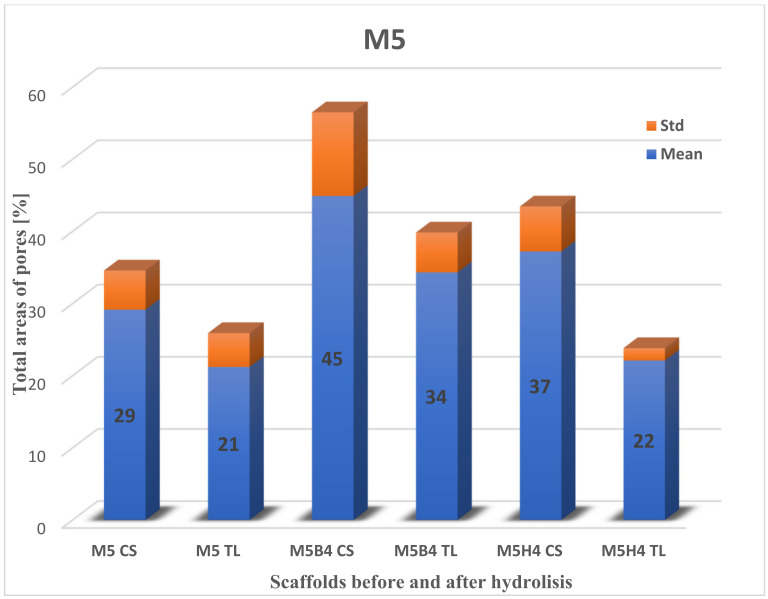
The total area of pores to the whole SEM image size for cross-section (CS) and the top layer (TL) of the M4 scaffold. B—degradation in SBF; H—degradation in HBSS.

**Figure 11 molecules-28-03195-f011:**
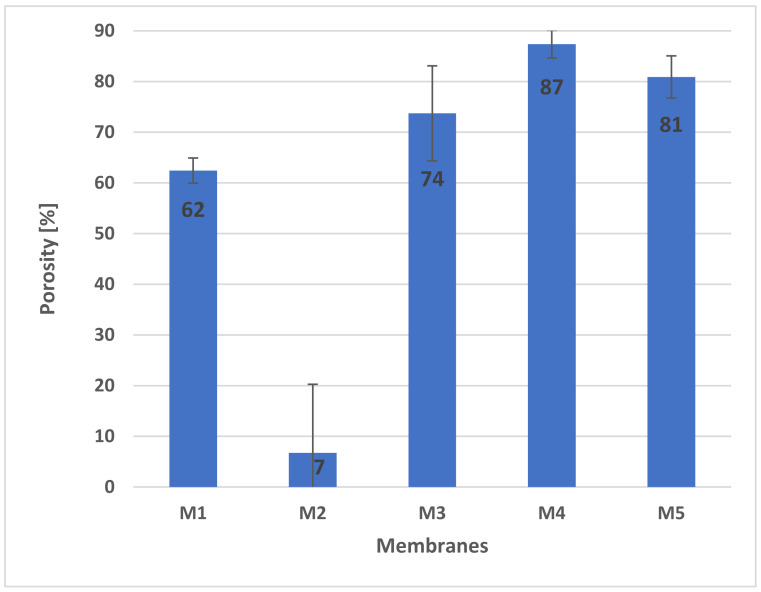
Porosity of membranes before degradation.

**Figure 12 molecules-28-03195-f012:**
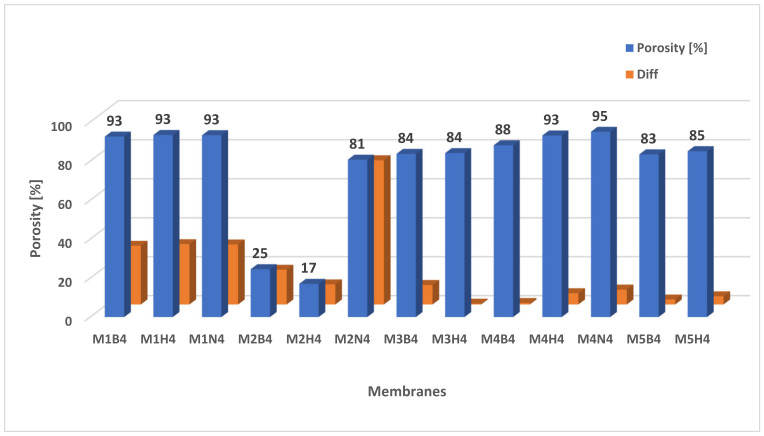
The porosity of membranes after degradation and difference (Diff) between membranes before and after hydrolysis.

**Figure 13 molecules-28-03195-f013:**
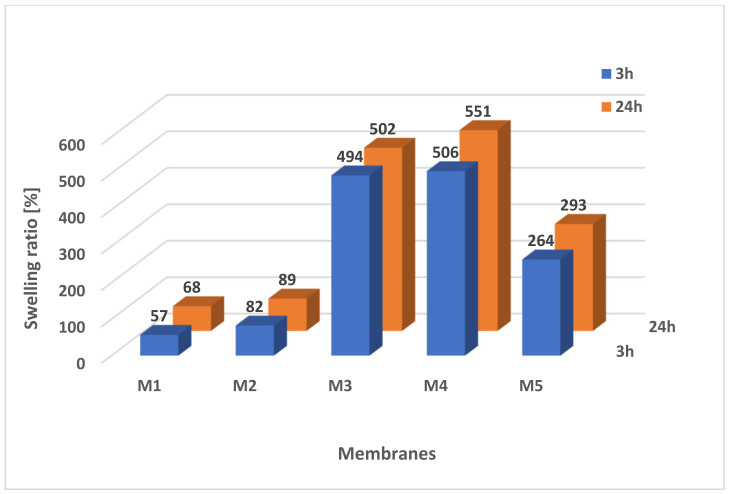
Swelling ratio of membranes in PBS after 3 and 24 h.

**Figure 14 molecules-28-03195-f014:**
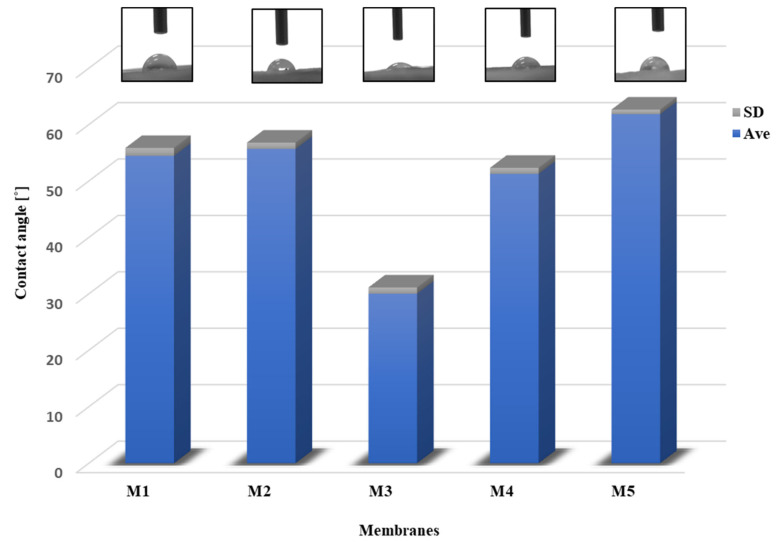
Static water contact angle of membranes. Above the diagram, photos of the surface of the membranes with a water droplet were presented.

**Figure 15 molecules-28-03195-f015:**
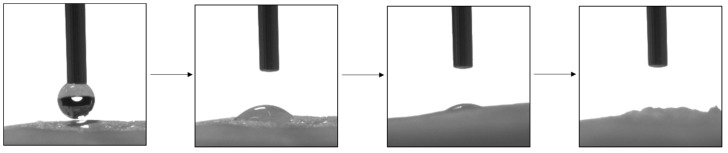
The surface of the M5 scaffold during water contact angle. The presented membrane was after 4 weeks of degradation in SBF fluid.

**Figure 16 molecules-28-03195-f016:**
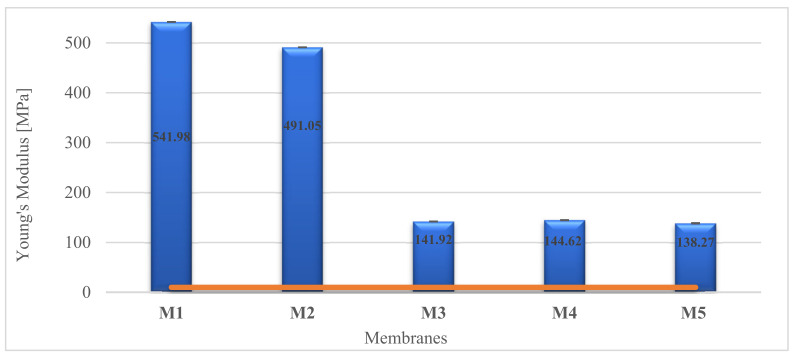
Mechanical properties of membranes. The red line shows the required Young’s Modulus for human cartilage.

**Figure 17 molecules-28-03195-f017:**
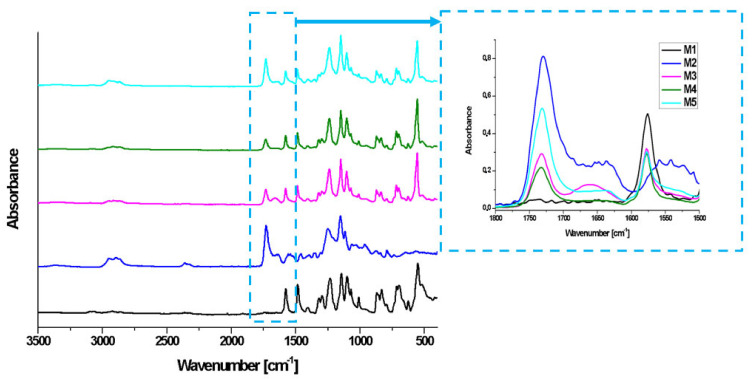
FT-IR analysis of membranes before degradation. In the blue square, the spectra were superimposed and magnified.

**Figure 18 molecules-28-03195-f018:**
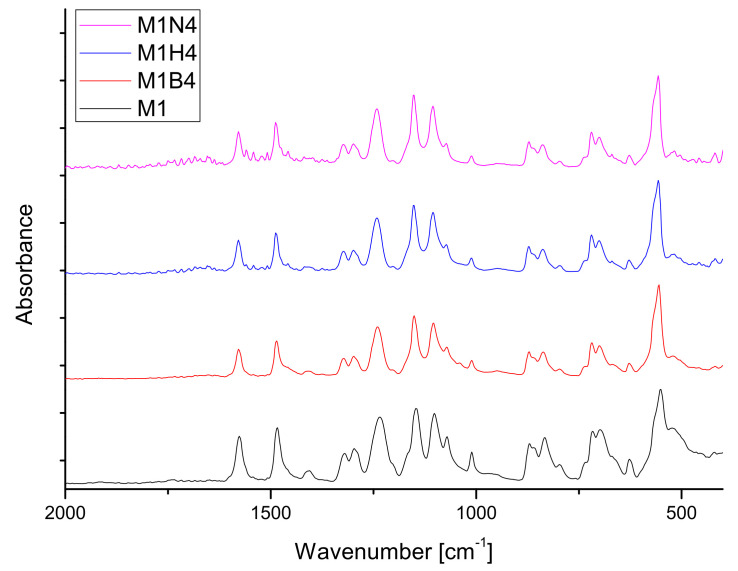
FT-IR spectra of M1 membrane before and after 4 weeks of degradation in each medium.

**Figure 19 molecules-28-03195-f019:**
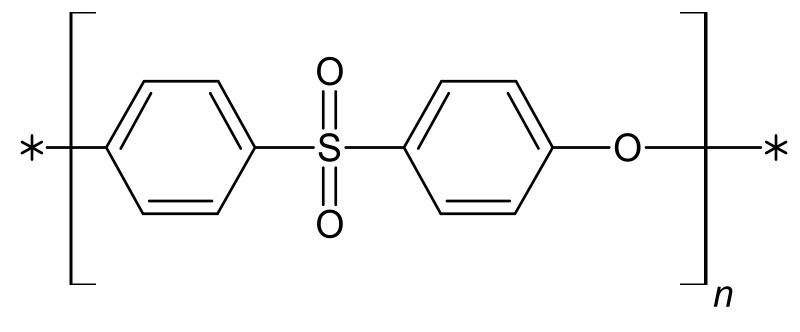
Structural formula of polyethersulfone (PES).

**Figure 20 molecules-28-03195-f020:**
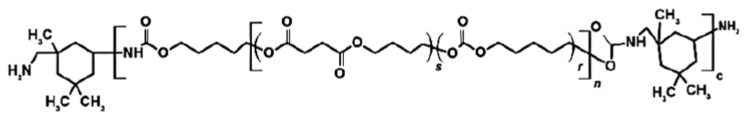
Chemical structure of polyurethane (PUR) [63].

**Figure 21 molecules-28-03195-f021:**
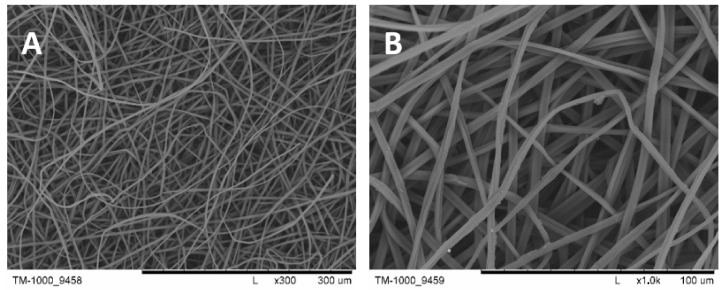
The photomicrographs of pork gelatin nonwovens obtained by an electrospinning method. Magnification: 300× (**A**) and 1000× (**B**).

**Figure 22 molecules-28-03195-f022:**
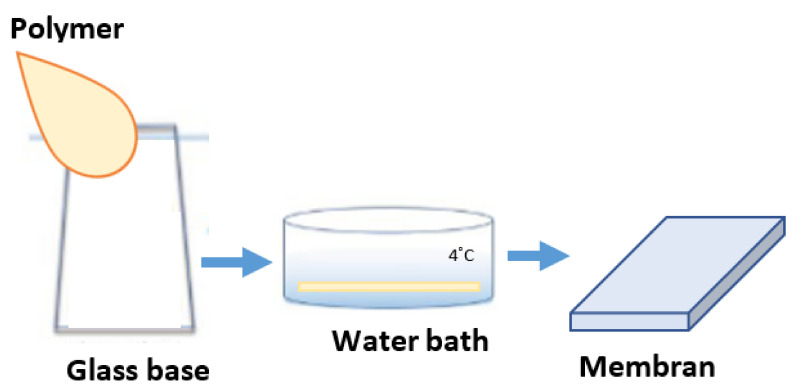
Scheme of preparation of references membrane M1–M2 by the wet-inversion method.

**Figure 23 molecules-28-03195-f023:**
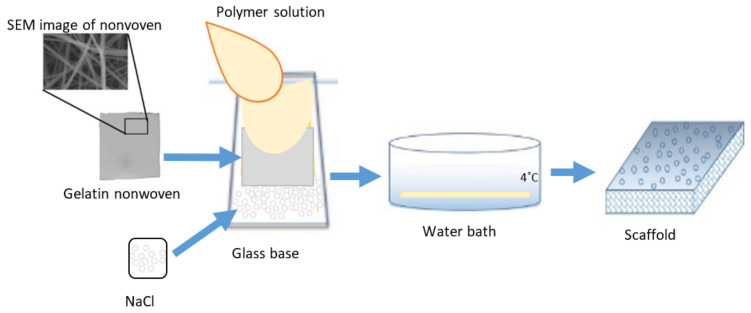
Schema of preparation of the M3–M5 scaffold.

**Table 1 molecules-28-03195-t001:** The pH of degradation media before and after 2 and 4 weeks of degradation.

Fluid	SBF					HBSS					NaOH					
Membranes	M1	M2	M3	M4	M5	M1	M2	M3	M4	M5	M1	M2	M3	M4	M5	
Initial pH	7.31					7.54					13.50					
Average pH after 2 weeks	7.42	7.47	7.67	7.05	7.30	7.39	7.26	7.10	7.58	6.98	13.80	13.89	13.99	13.78	13.84	
Average pH after 4 weeks	7.68	7.66	7.35	7.47	7.50	7.37	7.49	6.87	7.24	7.84	13.52	13.47	13.77	13.57	13.45	

**Table 2 molecules-28-03195-t002:** The difference in the total area of pores [%] for scaffolds before and after degradation in SBF and HBSS fluids.

Scaffold	Difference in HBSS Medium [%]	Difference in SBF Medium [%]
M3 CS	11.47	10.84
M3 TL	1.01	1.23
M4 CS	18.89	5.80
M4 TL	17.36	4.33
M5 CS	8.06	15.75
M5 TL	0.89	13.10

**Table 3 molecules-28-03195-t003:** Total areas of pores > 300 µm^2^ before and after degradation.

Scaffold	Total Area of Pores > 300 µm^2^ before and after Degradation		
	Before Degradation	After Degradation in HBSS	After Degradation in SBF
M3 CS	17.08 ± 5.61	32.11 ± 9.11	31.39 ± 6.26
M4 CS	20.66 ± 11.22	44.75 ± 10.46	24.82 ± 13.68
M5 CS	16.51 ± 7.72	27.33 ± 7.72	36.50 ± 9.12

**Table 4 molecules-28-03195-t004:** Weight ratios of PES and PUR polymers in the M3–M5 scaffolds.

Scaffold	PES:PUR Weight Ratios
M3	1:1
M4	2:1
M5	1:2

**Table 5 molecules-28-03195-t005:** Size classes of pores evaluated by MeMoExplorer™ Software.

j	1	2	3	4	5	6	7	8
Size µm^2^	0–3	3–8	8–20	20–80	80–100	100–150	150–300	>300

## Data Availability

Not applicable.

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
