# Peer review of "Scaffolds for Cartilage Tissue Engineering from a Blend of Polyethersulfone and Polyurethane Polymers"

_molecules, 2023, doi:10.3390/molecules28073195_

Round 1
Reviewer 1 Report
The M. Wasyłeczko with colleagues fabricated scaffolds for cartilage tissue engineering and further studied properties of scaffolds in different conditions.
The manuscript may be considered for publishing after a major revision and addressing the following comments:
1. English and grammatical errors should be rectified during the revision of the paper.
2. The Abstract should be rewritten in a better and clear way. The authors paid more attention on what they did. Instead, they should strengthen the introduction about their results.
3. The introduction is a fragment of a literature review. Authors should more clearly describe problems and state the need for a topic. What is the purpose of this research?
4. Text in introduction (lines 141-144) should be moved to the methods section.
5. The authors should introduce any of the abbreviation in the manuscript before using it (example lines 125, 138, etc.).
6. Authors need to rewrite the experimental section to make it easier to understand.
7. The used devices (lines 170-179) should be mentioned in the description of the methods.
8. The quality of figures 2, 3, 5-7, 18 should be improved.
9. The information about the composition of buffers and human plasma (table 3), SEM images after degradation (table 5), figure 20 should be placed in supplemental materials.
10. What is the size of the bar for the SEM images (Tables 4 and 5)?
11. What chemical reactions occur at degradation of the studied materials? Why does the pH of solutions change depending on the time of incubation?
12. Authors should more clearly describe the results of the effect of size and number of pores on degradation of scaffolds?
13. How do you explain the difference of scaffold degradation in HBSS and SBF?
14. Which samples (dry or wet) did you used for the mechanical test?
15. The discussion of the obtained results and their comparison with the results of other researchers poorly describe in manuscript.
Reviewer 2 Report
The work is devoted to obtaining and studying the properties of materials based on PES, PUR and their mixtures of various compositions. Such materials are indeed important for various areas of modern medicine.
In my opinion, the work presents a large number of results of various methods for studying samples, however, these results are not discussed in any way from the point of view of the objects of study themselves and the nature of changes in their properties, which makes it difficult to assess the completeness of the discussion of experimental data and the level of the study as a whole. The authors should significantly improve this component of their research. I believe that the presented studies are of interest to the readers of the Molecules journal and may be published after major revisions.
I also have some questions and comments:
1. The title of the study should be changed. The study is not a mini review. The introduction includes a traditional literature review, which is typical for all research articles.
2. What is the reason for the authors' choice of pore precursors used in the work? How was the completeness of pore precursors removal controlled?
3.The Materials and Methods section should be formatted according to the requirements of the Molecules Journal (MDPI).
4. Question on the methodology of the experiment. Why were gelatin and NaCl not used in the preparation of reference samples (M1 and M2)? Perhaps, the authors should include in the experimental part a graphical scheme of obtaining of experimental samples.
5. What determines the choice of media for membrane degradation experiments? In the work, the authors use two types of media (SBF and HBSS), which have almost identical salt composition. At the same time, the SBF solution is mainly used to assess the growth of the calcium phosphate layer on the implant surface, which the authors did not pay attention to in their study.
6. Mechanical properties. Why do the authors not provide stress-strain curves for samples, but only numerical values of Young's modulus? The curves would provide other valuable information about the mechanical characteristics of the samples.
7. Lines 547-549. The method of IR spectroscopy does not fully allow reliable confirmation of changes in the chemical composition of membranes during degradation, since they can occur mainly on their surface. Such changes should be investigated by other methods, for example, by the XPS method.
Reviewer 3 Report
1. Please end your abstract with a "take-home" message.
2. Line 23-25, the authors needs to giving the reverence for each mentioned aspect “…. such as joint prostheses [1], kidneys [2], heart valve [3], and other organs [4]”. Do not just fill a big group as present form like [1-8]” without referring specific for what. Also, for joint prosthesis, the authors suggesting the relevant reference that needs to adopted as follows: Computational Contact Pressure Prediction of CoCrMo, SS 316L and Ti6Al4V Femoral Head against UHMWPE Acetabular Cup under Gait Cycle. J. Funct. Biomater. 2022, 13, 64. https://doi.org/10.3390/jfb13020064
3. Put the keywords in a new order based on alphabetical order.
4. Abbreviation as a keyword is not recommended and encouraged to be changed become a stand for its abbreviation.
5. The Reviewer do not see the novel in the present article. My examination revealed that several similar previous publications appear to appropriately address the issues you have brought up in the current submission. Please emphasize it more advance in the introduction section if there are any more truly something really new.
6. Previous review related needs to explain in the introduction section consisting of their work, their novelty, and their limitations to show the research gaps that intend to be filled in the present review.
7. I am suggesting to transform the present review into systematic review and meta-analysis following PRISMA 2020 to giving more significant impact to the tissue engineering field.
Round 2
Reviewer 1 Report
The authors modified the manuscript taking into account my comments. The manuscript has been accepted for publication in Molecules.
Author Response
Dear Reviewer 1,
Thank you for your comments, suggestions, and tips. I appreciate the time and effort that you have dedicated to providing your valuable feedback on my manuscript.
On behalf of all authors,
Yours sincerely,
Monika Wasyłeczko
Reviewer 2 Report
The reviewer studied the revised version of the manuscript. The authors took into account all of my questions and comments and improved their manuscript in general. In my opinion, the manuscript can be accepted into Molecules journal in the present form.
Author Response
Dear Reviewer 2,
Thank you for your comments, suggestions, and tips. I appreciate the time and effort that you have dedicated to providing your valuable feedback on my manuscript.
On behalf of all authors,
Yours sincerely,
Monika Wasyłeczko
Reviewer 3 Report
Reviewers greatly appreciate the efforts that have been made by the author to improve the quality of their articles after peer review. I reread the author's manuscript and further reviewed the changes made along with the responses from previous reviewers' comments. Unfortunately, the authors failed to make some of the substantial improvements they should have made making this article not of decent quality with biased, not cutting-edge updates on the research topic outlined. In addition, the author also failed to address the previous reviewer's comments, especially on comments number 2 (not incorporating the literature), 5 (lack of novel), 6 (not captured state of the art), and 7 (systematic review and meta-analysis transformation not considered). Thank you very much for the opportunity to read the author's current work.
